# Gastro-Resistant Microparticles Produced by Spray-Drying as Controlled Release Systems for Liposoluble Vitamins

**DOI:** 10.3390/pharmaceutics14071480

**Published:** 2022-07-15

**Authors:** Francesca Terracina, Roberto Caruana, Francesco Paolo Bonomo, Francesco Montalbano, Mariano Licciardi

**Affiliations:** 1Dipartimento di Scienze e Tecnologie Biologiche Chimiche e Farmaceutiche (STEBICEF), Università degli Studi di Palermo, 90123 Palermo, Italy; francesca.terracina@unipa.it; 2Technology Scientific S.r.l., Viale delle Scienze, Edificio 18, 90128 Palermo, Italy; r.caruana@tecscien.com (R.C.); f.montalbano@tecscien.com (F.M.); 3Advanced Technologies Network Center (ATeN Center), Università degli Studi di Palermo, 90100 Palermo, Italy; francescopaolo.bonomo@unipa.it

**Keywords:** gastro-resistant microparticles, vitamins, spray-drying, whey proteins, Eudraguard, cellulose acetate phthalate

## Abstract

In the present study, gastro-resistant microparticles (MPs) were produced using the spray-drying technique as controlled-release systems for some model liposoluble vitamins, including retinyl-palmitate, retinyl-acetate, β-carotene, cholecalciferol and α-tocopherol. The gastroprotective action of three different gastro-resistant excipients, the anionic methacrylic copolymer (Eudraguard^®®^ Biotic, E1207), the cellulose acetate phthalate (CAP) and whey proteins (WPs), was compared. The latter was used to produce a novel delivery system manufactured with only food-derived components, such as milk, and showed several improvements over the two synthetic gastro-resistant agents. Scanning electron microscopy (SEM) images showed a quite homogeneous spherical shape of all microparticle batches, with an average diameter between 7 and 15 μm. FTIR analysis was used to evaluate the effective incorporation of vitamins within the microparticles and the absence of any degradation to the components of the formulation. The comparison graphs of differential scanning calorimetry (DSC) confirmed that the spray drying technique generates a solid in which the physical interactions between the excipients and the vitamins are very strong. Release studies showed a prominent pH-controlled release and partially a delayed-release profile. Ex vivo permeation studies of retinyl palmitate, retinyl acetate and α-tocopherol revealed greater transmucosal permeation capacity for microparticles produced with the WPs and milk.

## 1. Introduction

Vitamins are among the most important nutrients necessary for human health since they act as regulators of the organism’s development. The physiological functions of vitamins are highly specific, and for this reason, they are required only in small quantities in the diet. Our body is unable to produce them, so the best way to reach the daily requirement, according to the Recommended Dietary Allowances (RDA), is to eat foods that contain them. Conditions of a partial deficiency of vitamins, due to reduced dietary intake, reduced absorption, or increased need, lead to states of hypovitaminosis that manifest themselves through different types of pathologies [1]. In these cases, oral intake of vitamins is commonly recommended beyond the normal diet.

Vitamins are classified as either fat-soluble or water-soluble. Water-soluble vitamins (WSV) have one or more polar or ionizable groups (vitamin C, thiamine, riboflavin, niacin, pyridoxine, pantothenic acid, biotin, folic acid, and vitamin B12); the fat-soluble vitamins (FSV) have mainly aromatic and aliphatic characters and include vitamin A, E, K and D. Water-soluble vitamins easily dissolve in water and generally are excreted readily from the body; so, as they are not stored readily, continuous intake is important. Lipophilic vitamins, in particular, do not readily dissolve in the gastrointestinal (GI) tract and are absorbed preferentially via the intestinal tract and then reach the bloodstream [2].

Both WSV and FSV are important in our body for the prevention of a number of pathologies, particularly those with an inflammatory and oxidative basis, such as neurodegenerative diseases (e.g., Parkinson’s disease, Alzheimer’s disease, Huntington’s disease, multiple sclerosis, amyotrophic lateral sclerosis and prion disease) [3].

Moreover, recent studies have highlighted a bidirectional relationship between gut microbiota (GM) and FSV. FSV influence the composition and function of the GM through the regulation of the immune response and inflammatory pathways. On the other hand, GM and dysbiosis influence the status, metabolism and functions of FSV [4].

Vitamin A or retinol belongs to a group of unsaturated nutritional organic compounds and is usually taken up as retinyl esters, retinyl acetates, or pro-vitamin A molecules such as β-carotene. These compounds have several potential human health benefits, e.g., decreased risk of some cancers, cardiovascular diseases, age-related macular degeneration, cataracts, increased immune response and an essential role in vision and cell differentiation in humans [5,6]. According to UNICEF, vitamin A deficiency is the leading cause of preventable childhood mortality [7].

Vitamin D is one of the most important micronutrient supplements that leads to improving overall human health. The term “Vitamin D” refers to both ergocalciferol (vitamin D2) and cholecalciferol (vitamin D3) [8]. Cholecalciferol has an increasingly acknowledged role in the development of several autoimmune diseases, including type 1 diabetes, multiple sclerosis, rheumatoid arthritis and inflammatory bowel disease [9].

Vitamin E exists in eight different forms, four tocopherols and four tocotrienols. α-tocopherol is a well-known and well-studied component of vitamin E. Being an antioxidant, vitamin E protects the membrane fats from oxidative damage and maintains cellular functioning [10,11].

Nowadays, it is very common to consume such vitamins as supplements, despite the efficacy of the above lipophilic vitamins being often compromised by their poor water solubility, rapid degradation, and low absorption, resulting in low bioavailability. To overcome these problems, researchers have focused on promising delivery technologies, such as microparticles, which allow improvement of the dissolution and permeation of these active ingredients according to a gastro-resistant profile. On the other hand, controlled vitamin release may reduce potential hypervitaminosis syndromes, side effects or frequent administration [12].

Microencapsulation techniques such as spray-drying can be the optimal solution. Spray-drying is a drying technique that has met with ever-increasing success in recent years, which can be attributed to the considerable technological advantages that it can guarantee to the final product, including reduction of the particle size of the dried product; uniformity of content; efficiency of the microencapsulation process; and improvement of the pharmacokinetic profile of hydrophobic drugs [13,14].

The use of this drying technique, assisted by preformulation studies on the excipients to be used, also allows obtaining modified and/or controlled-release systems, particularly as in this study, one of the typical gastro-resistant dosage forms. These are systems that direct vitamins to a specific release site, such as the enteric tract, thus overcoming one of the major limitations of the oral route of administration: the inactivation or degradation of drugs due to gastric acidity [15].

In the present work, using the spray-drying technique, gastro-resistant microparticles (MPs) were produced containing the model vitamins, such as retinyl palmitate, retinyl acetate, β-carotene, cholecalciferol, α-tocopherol and the gastro-protective action of three different gastro-resistant agents were compared: two commercial agents, the anionic methacrylic copolymer (Eudraguard^®®^ Biotic) and cellulose acetate phthalate (CAP), and a food-derived component, such as whey proteins (WPs) [16].

Eudraguard Biotic is an excipient system based on an anionic methacrylic copolymer belonging to the Eudragit family and used as a highly flexible film-forming material, capable of protecting the solid dosage forms from gastric acidity, preventing their degradation in the stomach and promoting the release of the active ingredient in the intestinal region. The difference between Eudraguard Biotic and derivatives of traditional Eudragit is that it has high biocompatibility orally, being included in the list of “EU Approved additives and E Numbers” with the label E1207. CAP is a pH-sensitive derivative of cellulose, suitable for coating tablets, capsules or granules in the pharmaceutical sector. It is insoluble in contact with gastric fluids, while readily dissolving in the neutral or basic environment of the small intestine. WPs are food-derived ingredients produced from milk, acknowledged as Generally Recognized As Safe (GRAS) substances, and successfully proposed as a biocompatible carrier for bioactive molecules [17,18].

These features guarantee a micro delivery system able to increase the percentage of administered dose reaching the absorption site and better suit the concept of all-natural formulation [19,20].

This study aimed to demonstrate that proper formulations of vitamin-loaded MPs, produced by spray-drying, may improve the oral absorption of the loaded vitamins, taking advantage of the formulation composition, such as pH-dependent solubility, site-selective release, drying characteristics and higher stability.

## 2. Materials and Methods

### 2.1. Materials

Retinyl palmitate, β-carotene, and α-tocopherol were purchased from A.C.E.F., Italy; cholecalciferol, mannitol, n-octanol, ethanol and CAP were purchased from Aldrich, Italy; retinyl acetate was purchased from Pharmorgana, Germany; gum arabic and soy lecithin were purchased from Galeno, Italy; WPs were purchased from Bulk Powders, England; Eudraguard biotic was purchased from Evonik, Essen, Germany; and HCl was purchased from VWR.

### 2.2. Methods

The particle’s size and morphology were measured by scanning electron microscopy (SEM) imaging technique, using a Phenom ProXSEM. Each sample was deposited onto a carbon-coated steel stub and dried under vacuum (0.1 Torr) before analysis. All of the SEM analyses were performed at 25.0 °C ± 0.1 °C. The average diameter (d) ± standard deviation (SD) (mean ± SD, *n* = 3) of the microparticles was determined from the mean value of 100 measurements using ImageJ (Madison, WI, USA, version 1.46 v).

Fourier-transform infrared spectroscopy (FTIR) samples were prepared in KBr tablets and analyses were performed with Jasco FTIR-6000 in the range 4000–400 cm^−1^ and 128 scans. Vitamins, pure excipients, physical mixtures of them and final micro delivery systems were analyzed.

Differential scanning calorimetry (DSC) analyses were performed with Setaram DSC131 EVO, at a heating rate of 20 °C/min and 120 µL using aluminum non-hermetic crucibles. Pure active ingredients, pure excipients, physical mixtures of them and final micro delivery systems were analyzed.

Thermogravimetric analyses (TGAs) were performed with Labsys TGA-DSC Setaram, at a heating rate of 5 °C/min between 25 and 550 °C using alumina crucibles. The water content ± SD (mean ± SD, *n* = 3) was calculated by the instrument software in the range 25–120 °C. The analysis was performed immediately after the production of each batch of microparticles and subsequently 6 months after their production.

#### 2.2.1. Preparation of the Gastro-Resistant Microparticle Formulations

The micro delivery systems were prepared with the Mini Spray-Dryer Buchi B290 (Buchi, Germany). The spray drying process was performed according to the following parameters: inlet T: 130 °C; outlet T: 70–73 °C; aspiration: 100%; feed pump: 15% (4.5 mL/min); atomizer nozzle: 0.7 mm; used gas: compressed air. The manufacture started with the preparation of the feed dispersion, which was a suspension containing the active molecules and the excipients forming the micro delivery systems.

To prepare Eudraguard Biotic gastro-resistant microparticles (batches A1, B1, C1, D1, E1), 2 g of gum arabic (stabilizing agent), 2.5 g of mannitol (diluent agent) and 0.25 g of soy lecithin (emulsifying agent) were dispersed in 45 mL of distilled water under continuous stirring at 40 °C for 30 min, then the feed dispersion was cooled at 30 °C, and a specific amount of the selected vitamin (0.5 g retinyl palmitate, 0.5 g beta-carotene, 0.5 g retinyl acetate, 0.1 g cholecalciferol, or 0.25 g α-tocopherol) solubilized in 5 mL of ethanol was added, and the dispersion was stirred for 15 min. Finally, 2 mL of a 30% w/v solution of Eudraguard Biotic (corresponding to 0.6 g) were added and stirred for 15 min.

To prepare CAP gastro-resistant microparticles (batches A2, B2, C2, D2, E2), 0.25 g of CAP were solubilized in 45 mL of distilled water, maintaining the pH near 9 (adding NaOH 0.1 M) at 40 °C under continuous stirring for 2 h. Then 2 g of gum arabic (stabilizing agent), 2.5 g of mannitol (diluent agent) and 0.25 g of soy lecithin (emulsifying agent) were added, maintaining 40 °C and stirring for 30 min. The feed dispersion was cooled at 30 °C, and a specific amount of the selected vitamin (0.5 g retinyl palmitate, 0.5 g beta-carotene, 0.5 g retinyl acetate, 0.1 g cholecalciferol, or 0.25 g α-tocopherol) solubilized in 5 mL of ethanol was added, and the dispersion was stirred for 15 min.

To prepare WP gastro-resistant microparticles (batches A3, B3, C3, D3, E3), 2.5 g of WPs were dispersed in 50 mL of whole milk (total dry residue in 100 mL: 3.6 g of fat, 4.8 g of carbohydrates, 3.2 g of protein, 0.13 g of salt and 0.12 g of calcium) under continuous stirring at 25 °C for 30 min; then, a specific amount of the selected vitamin (0.5 g retinyl palmitate, 0.5 g beta-carotene, 0.5 g retinyl acetate, 0.1 g cholecalciferol, or 0.25 g α-tocopherol) was added and the dispersion was stirred for other 30 min.

The formulation compositions of all of the batches subjected to the spray drying process are summarized in Appendix A.

According to the conditions reported above, the feeds were nebulized and dried by the spray dryer. After the microparticle production, the percentage yield of the process (% Y) ± SD (mean ± SD, *n* = 3) was calculated as:$$\%Y=\frac{\mathrm{Mass}\;\mathrm{of}\;\mathrm{excipients}\;\mathrm{and}\;\mathrm{active}\;\mathrm{ingredient}\;\mathrm{in}\;\mathrm{the}\;\mathrm{formulation}}{\mathrm{mass}\;\mathrm{of}\;\mathrm{microparticles}\;\mathrm{obtained}}\times 100$$

#### 2.2.2. Drug Loading Evaluation

The drug loading (DL%) ± SD (mean ± SD, *n* = 3) was calculated according to the formula below, and the amount of vitamin actually encapsulated in each micro delivery system was assessed spectrophotometrically using a UV Jasco V760 spectrophotometer (cuvette 1 mL, optical path 10 mm).
$$\%\mathrm{DL}=\frac{\mathrm{Mass}\;\mathrm{of}\;\mathrm{active}\;\mathrm{ingredient}\;\mathrm{in}\;\mathrm{microparticles}}{\mathrm{mass}\;\mathrm{of}\;\mathrm{microparticles}}\times 100$$

The encapsulation efficiency (%EE) ± SD (mean ± SD, *n* = 3) was determined by accounting for the encapsulated vitamins within their respective micro delivery systems and the amount introduced in the feed of the spray dryer.
$$\%\mathrm{EE}=\frac{\mathrm{Mass}\;\mathrm{of}\;\mathrm{active}\;\mathrm{ingredient}\;\mathrm{in}\;\mathrm{microparticles}}{\mathrm{mass}\;\mathrm{of}\;\mathrm{active}\;\mathrm{in}\;\mathrm{the}\;\mathrm{feed}}\times 100$$

All of the experiments were carried out in triplicate.

#### 2.2.3. In Vitro Dissolution Studies

Each micro delivery system and each relatively pure vitamin were placed inside two filter paper sachets and immersed in a beaker containing the bi-phasic medium, 15 mL of n-octanol and 15 mL of 0.1 M HCl. The system was maintained at pH = 1, under stirring and constant temperature (100 rpm, 37 °C) for 2 h. After 2 h, the pH of the system was increased up to 6.8 by adding NaOH 1 M to the aqueous phase (to simulate the intestinal environment). This pH was maintained for the following 6 h. One ml sample of the upper organic phase (n-octanol) was taken at regular time intervals, immediately read with the spectrophotometer UV Jasco V-760 (cuvette 1 mL, optical path 10 mm) and poured into a beaker.

#### 2.2.4. Ex-Vivo Permeation Studies

Permeation studies were performed with Franz cells and porcine colon mucosae within an orbiting incubator at 37 °C (±1 °C). Five ml of n-octanol were placed in the acceptor compartment, and 1.5 mL of 0.1 M HCl in the donor compartment. Two-hundred microliter samples from the acceptor compartment were withdrawn every 60 min for 8 h, immediately diluted up to 1 mL with n-octanol, and read with the spectrophotometer UV Jasco V-760 (cuvette 1 mL, optical path 10 mm). Each volume withdrawn for sampling was replaced by an equal amount of solvent. The studies were carried out in triplicate, and graphs of the permeation curves were generated, reporting permeated vitamin in μg vs. time.

#### 2.2.5. Statistical Analysis

Results obtained from multiple samples were expressed as mean ± standard deviation (SD) (*n* = 3). Statistical differences were analyzed by Student’s *t*-Test. *p*-value < 0.05 was defined as the level of statistical significance.

## 3. Results and Discussion

Spray-drying is an advanced micronization process that produces improved supplement formulations with higher drug dissolution, absorption, and therapeutic efficacy when compared to traditional dosage forms [13,15,21].

Each micro delivery system was developed and optimized with the following active ingredients: retinyl palmitate (A), β-carotene (B), retinyl acetate (C), cholecalciferol (D), and α-tocopherol (E) (Figure 1).

Several batches of each micro delivery system were produced to optimize the spray drying process, yields and DL [21,22].

Three formulations for each vitamin were chosen in this study: batches A1, B1, C1, D1, and E1 containing the gastro-resistant excipient Eudraguard Biotic; batches A2, B2, C2, D2, and E2 containing the gastro-resistant excipient CAP; batches A3, B3, C3, D3, and E3 containing WPs (Figure 2).

The process yield ranged from 43% to 66%, and the DLs of the active ingredients ranged between 0.5% and 3.5% as listed in Table 1. These results agree with the literature regarding the use of this microparticle production technique, and it is believed that optimal process yield values are between 30–70% when using the mini spray dryer Buchi B-290 [21,23,24]. The encapsulation efficiency (EE) of all batches containing the same vitamin was not homogeneous. Actually, a relationship between DL and EE was not evidenced.

The characterization studies of produced microparticles included SEM analysis firstly, with the aim of observing the surface characteristics of the microparticles produced and their average diameter (μm) (Table 2).

SEM images (Figure 3) revealed quite homogeneous spherical shapes in all microparticle batches. In particular, batches 1 (average diameter between 9 and 14 μm) and 2 (average diameter between 9 and 12 μm) of the different active ingredients appeared to have a more spherical homogeneous morphology and a smaller average diameter size range, probably due to the presence of different excipients, such as mannitol, gum arabic, and Eudraguard Biotic, which improved their surface characteristics. On the other hand, batch 3 (average diameter between 7 and 15 μm) showed a more non-homogeneous morphology with several collapsed microparticles, probably due to the lack of excipients capable of guaranteeing these characteristics. In spite of these differences, the vitamins were encapsulated within the MPs, and no correlation was noted between these surface characteristics and the dissolution and/or permeation profiles of these systems.

The FTIR analysis was used to evaluate the effective incorporation of vitamins within the microparticles, as well as the absence of any degradation to the components of the formulation (Appendix A). In pure crystalline form, vitamins were analyzed to identify the possible presence of drug–excipient interactions, occurring between specific chemical groups within their structures.

Table 3 summarizes the peak assignment of each pure vitamin and the respective peaks found in each formulation. The identified peaks found relevant correspondence in the literature, and no particular signs of vitamin degradation were detected.

DSC analysis was used to evaluate if the production process of microparticles allows the creation of chemical–physical interactions between the excipients and the active ingredient used in each batch produced using the spray-drying technique [25]. In the DSC graphs in Appendix A, the difference in the thermal behavior of each microparticle batch was compared to the thermal behavior of the simple physical dry mixture of all the of components used in the respective formulation, and with the pure component as well. In all comparisons of DSC graphs, it was noticed that the typical melting/degradation peaks of each pure vitamin, still visible in the physical mixture, disappeared in the DSC graph of the batch of microparticles produced by spray-drying, with the appearance of a new peak at a temperature different from that of the individual components and characteristic only of the microparticles. This suggests that the drying process with the spray dryer generates a solid in which the physical interactions between the excipients and the incorporated vitamin are very strong.
pharmaceutics-14-01480-t003_Table 3Table 3FTIR peak assignment for each vitamin and MP batch: A1 (retinyl palmitate and Eudraguard Biotic), A2 (retinyl palmitate and CAP), A3 (retinyl palmitate and WPs), B1 (β-carotene and Eudraguard Biotic), B2 (β-carotene and CAP), B3 (β-carotene and WPs), C1 (retinyl acetate and Eudraguard Biotic), C2 (retinyl acetate and CAP), C3 (retinyl acetate and WPs), D1 (cholecalciferol and Eudraguard Biotic), D2 (cholecalciferol and CAP), D3 (cholecalciferol and WPs), E1 (α-tocopherol and Eudraguard Biotic), E2 (α-tocopherol and CAP), E3 (α-tocopherol and WPs).**Retinyl Palmitate cm^−1^ [26]**



**2922.11**A1 cm^−1^A2 cm^−1^A3 cm^−1^Stretching C-H**1734.66**2922.112922.112922.11Stretching C=O**9632.65**1734.171734.171744.76Bending =CH**β-carotene cm^−1^ [27,28]**B1 cm^−1^B2 cm^−1^B3 cm^−1^
**3440.39**

3400.85Stretching =CH**2935.61**2935.132935.132935.13Stretching =CH**1721.64**1735.821735.821721.64Stretching C=C**1450.69**1449.731449.731450.69Bending C-H**Retinyl acetate cm^−1^ [29]**C1 cm^−1^C2 cm^−1^C3 cm^−1^
**3444.24**


Stretching =CH**2926.45**2935.612936.092924.52Stretching C-H**1624.73**

1653.18Stretching C=O**1436.71**1456.981457.44
Stretching C=C**Cholecalciferol cm^−1^ [30]**D1 cm^−1^D2 cm^−1^D3 cm^−1^
**3309.73**


Stretching O-H**3077.35**


Stretching C-H**2933.68**2935.132935.132933.68Stretching -CH_2_**1645.91**1645.911645.911645.91Stretching C=C**α-tocopherol cm^−1^ [31,32]**E1 cm^−1^E2 cm^−1^E3 cm^−1^
**3632.27**


Stretching =OH**1420.32**1463.221462.74
Bending -CH_3_**1087.17**1087.171087.661070Plane Bending phenyl


In particular, for the retinyl palmitate microparticle formulations A1 and A2, an exothermic peak was observed in the range of 150–160 °C, attributable to its oxidation [26]. Similar behavior was shown for β-carotene (microparticle formulations B1 and B2) differently from the characteristic melting peak of the pure vitamin around 170 °C [33]. Retinyl acetate (microparticle formulations C1 and C2), cholecalciferol (microparticle formulations D1 and D2) and α-tocopherol (microparticle formulations E1 and E2) instead had fusion peaks around 45 °C, 60 °C and 90 °C, respectively [32,34]. The Eudraguard Biotic and CAP thermograms showed an endothermic peak, respectively, at around 50 °C and 170 °C, attributable to the glass transition of the amorphous polymer (T_g_) [35,36]. All of the microparticle formulations containing WPs (A3, B3, C3, D3, E3) showed a single broad peak around 130 °C, slightly different from the characteristic peak of decomposition of whey proteins around 120 °C [17].

No unusual peaks of vitamin degradation were observed in the microparticle thermograms, indicating that a correct operating temperature was used during the production process, which preserved the integrity of the loaded active ingredients.

TGA analysis was performed on the batches of microparticles in order to evaluate the residual humidity in the solid product and hygroscopicity. All microparticle samples analyzed immediately after the production process showed an average residual percentage of water equal to about 2.5% by weight, as expected. TGA analyses were also repeated after storing the microparticle samples at a relative humidity of about 30–40%, 25 °C for 6 months after their production. The results obtained in terms of mass variation and percentage mass variation due to water loss of the samples (residual humidity content) are summarized in Table 4. Overall, the batches of microparticles showed a slight increase in residual humidity, but no visible degradation phenomena. Only batches with β-carotene (B1, B2, B3) had statistically different lower values compared with each of the other vitamins.

An in vitro dissolution study was carried out to highlight the differences in the release profiles of loaded active molecules, generated by the gastro-resistant ingredients encapsulated in the MPs, compared to the pure actives. The evaluation of the effect on the dissolution profile of the microparticles made it possible to appreciate the potential of the spray-drying technique to obtain delayed-release formulations (pH-sensitive/gastro-resistance release). In this regard, an in vitro model was used to simulate the distribution of the active molecules between the aqueous phase of the gastric fluid and the lipid phase of the biological membranes of the epithelial cells of the gastrointestinal wall. This in vitro model could also be approximated to an absorption pattern through the intestinal membrane. In this case, the lipid phase of the membranes was mimicked through the use of an organic solvent, n-octanol, in a biphasic system with an aqueous solution (Figure 4) [37].

The aqueous phase was maintained at pH 1 for the first two hours of the experiment in order to reproduce the gastric passage; subsequently, the pH was raised to 7.4 to simulate intestinal transit.

From the results obtained in these dissolution studies (Figure 5A–E), it was possible to observe that all of the batches of tested microparticles had dissolution profiles that were not triggered (or instantaneous), like that of the pure active ingredient, but modified (e.g., prolonged and/or pH-sensitive). In particular, all three batches of each active ingredient formulated with WPs and milk were those showing the best pH-sensitive release profiles.

In particular, in the case of retinyl palmitate, the percentage of the released vitamin from the formulations after 2 h was equal to 7% for batch A1, 35% for batch A2, and 57% for batch A3 and exceeded 65% in the case of the pure vitamin. After 2 h, the pH was changed, and an increase in the quantity of vitamin released by the systems was observed over time: for batch A1, a pH-controlled release was observed, passing from 7% at 2 h up to 100% at 8 h; batch A2 after the pH change immediately released all the incorporated vitamin; batch A3 showed a controlled and delayed-release profile, reaching 100% vitamin released at 8 h; the pure vitamin completed its dissolution after 4 h (Figure 5A).

In the case of β-carotene (Figure 5B), it was possible to observe that after about 4 h, the microparticles of batches B1 and B2 released 100% of the loaded vitamin. Pure β-carotene reached a maximum dissolution percentage of about 50% after 8 h. Although the two batches of microparticles did not show pH-sensitive behavior, it is important to underline that the two formulations guaranteed technological advantages to the microparticles produced, since the microencapsulated β-carotene was found to have greater solubility and dissolution rate than the pure vitamin. This result is probably due to its co-formulation with water-dispersible excipients, which improved its release profile compared to the pure active ingredient. Batch B3, on the other hand, showed a typical pH-sensitive profile: in the first 2 h, 13% of the active ingredient was released, and only after the pH change, an increase in the amount of released β-carotene was revealed.

There was no significant difference between the release profile of retinyl acetate from the C1 microparticles and the pure active ingredient; both showed a particularly immediate release profile. Instead, with batches C2 and C3, the above release profile was modified, obtaining a prolonged release system (Figure 5C).

The percentage of cholecalciferol released after 2 h was 34% for batch D1, 72% for batch D2, 25% for batch D3, and 45% for the pure vitamin. After the pH change, the quantity of released vitamin gradually increased, reaching up to 100% for batches D1 and D2. The pH-sensitive effect was more evident for batch D3, which slowed down the release of cholecalciferol, still releasing over 8 h (Figure 5D).

Even for formulations with α-tocopherol, it was batch E3 that had a slight pH-sensitive release profile: it was only after the pH jump that the greatest amount of encapsulated active ingredient was released gradually, reaching up to 60% at 8 h. The E1 and E2 systems instead showed an immediate release profile comparable to that of the pure active ingredient (Figure 5E).

Through the ex vivo permeation study, it was possible to evaluate the potential increased permeation of microencapsulated vitamins through the porcine intestinal mucosa used as a model tissue. To simulate transmucosal absorption, Franz’s vertical diffusion cells were used as an in vitro permeation model. The pH and temperature parameters were chosen so as to simulate those characteristics of the gastrointestinal passage of a formulation taken via the oral route. The porcine intestinal mucosa was placed between the two compartments as shown in Figure 6.

Regarding the batches of microparticles containing β-carotene and cholecalciferol, no permeation of vitamins through the porcine intestinal mucosa was observed, even in the pure form, as these active ingredients exploit in vivo transport systems that cannot be reproduced in vitro.

Actually, ex vivo permeation studies of retinyl palmitate (Figure 7A), retinyl acetate (Figure 7B) and α-tocopherol (Figure 7C) revealed greater transmucosal permeation capacity for microparticles produced with the WPs and milk.

Furthermore, the pH-sensitive release profiles of these batches, which had already been observed during the dissolution studies, were approximately confirmed. In addition, the obtained results unexpectedly showed that WPs can actually act as absorption enhancers, improving the bioavailability of lipid molecules, such as fat-soluble vitamins [17].

These results encourage further exploration of the potential of WPs as absorption promoters for molecules of pharmaceutical interest, also through in vivo bioavailability studies. In fact, this advantageous feature has already been recently demonstrated with active molecules of natural origin [17]. Furthermore, it will be interesting to understand if specific components of WPs, such as β-lactoglobulins, are directly responsible for these beneficial vehicle properties [38].

## 4. Conclusions

In the present study, three different gastro-resistant microparticle formulations were successfully produced using the spray-drying technique and characterized. In particular, the effect of the formulation composition was explored to control the release profiles of some model lipid vitamins, such as retinyl palmitate, retinyl acetate, β-carotene, cholecalciferol and α-tocopherol, by using three different gastro-resistant excipients—two synthetic, the anionic methacrylic copolymer (Eudraguard^®®^ Biotic, E1207) and CAP, and one natural, such as WPs. In vitro release results showed that the pH-dependent solubility of each formulation was strictly dependent not only on the kind of gastro-resistant excipient used, but also on the chemical–physical characteristics of the delivered active molecules. For example, the best pH-dependent release profiles proved to be A1 > B3 > D3 > D2 > A2 > E3. Otherwise, it seems to be established that WPs can act not only as gastro-resistant agents but also as absorption promoters for lipid molecules, as in the case of the vitamins used iFiguren this study. Actually, ex vivo permeation studies through porcine intestinal mucosa demonstrated that oral absorption of the lipid vitamins retinyl palmitate, retinyl acetate and α-tocopherol can be drastically improved when loaded into microparticles produced using WPs and milk by the spray-drying technique, as reported here. The obtained results suggest the achievement of important advantages in the pharmaceutical field, namely the possibility of using an encapsulation process for fat-soluble active ingredients using the spray-drying technique with a completely natural formulation composition, able to improve the release profile of the active ingredient and its absorption capacity, and increasing its bioavailability in vivo.

## 5. Patents

Patent n. IT2020000030992, SISTEMA PER IL RILASCIO CONTROLLATO DI PRINCIPI ATTIVI.

## Figures and Tables

**Figure 1 pharmaceutics-14-01480-f001:**
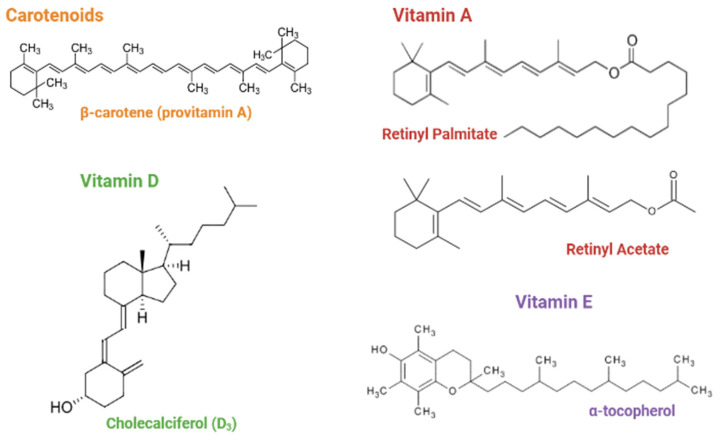
Chemical structures of the liposoluble vitamins discussed in this article.

**Figure 2 pharmaceutics-14-01480-f002:**
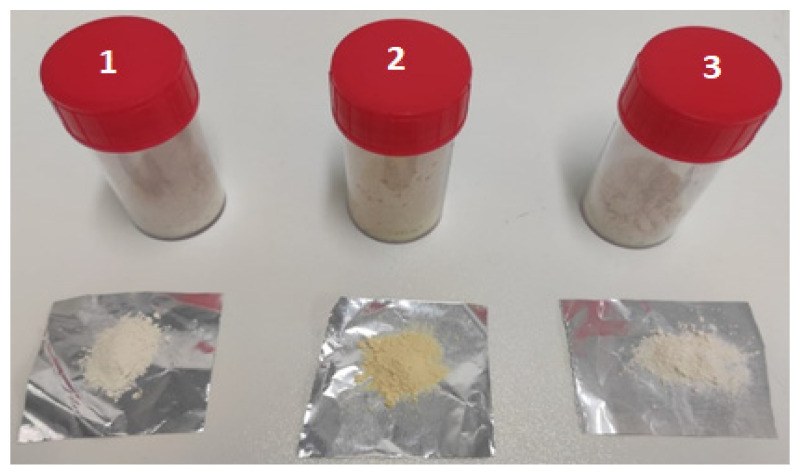
Macroscopic appearance of some microparticle samples obtained from the spray-drying process: 1 (batch A1); 2 (batch B2); 3 (batch C3).

**Figure 3 pharmaceutics-14-01480-f003:**
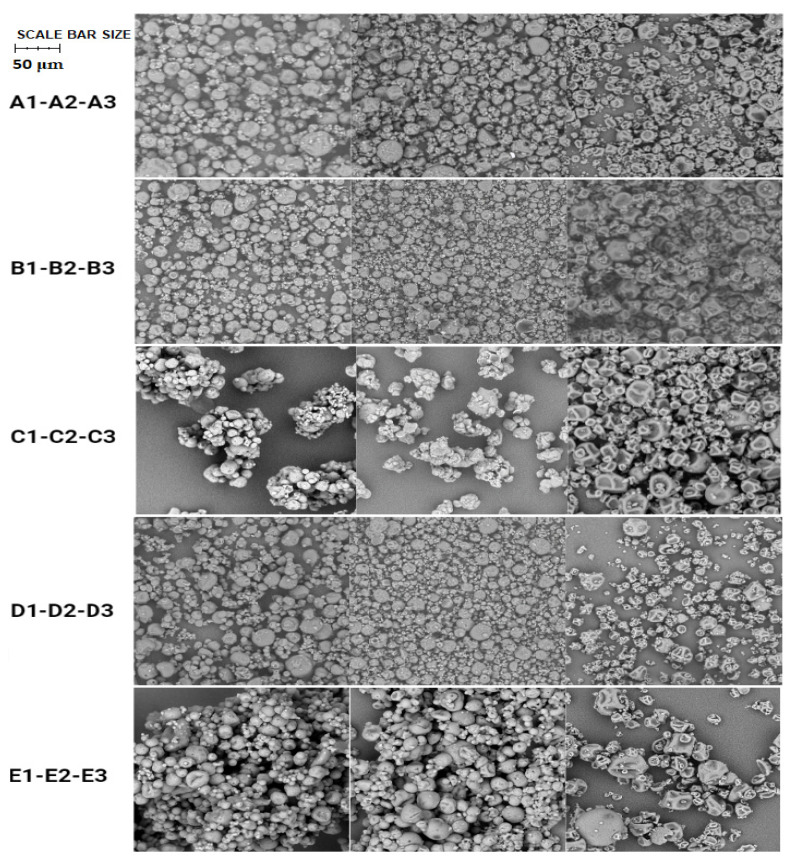
SEM images of the microparticle batches: A1 (retinyl palmitate and Eudraguard Biotic), A2 (retinyl palmitate and CAP), A3 (retinyl palmitate and WPs), B1 (β-carotene and Eudraguard Biotic), B2 (β-carotene and CAP), B3 (β-carotene and WPs), C1 (retinyl acetate and Eudraguard Biotic), C2 (retinyl acetate and CAP), C3 (retinyl acetate and WPs), D1 (cholecalciferol and Eudraguard Biotic), D2 (cholecalciferol and CAP), D3 (cholecalciferol and WPs), E1 (α-tocopherol and Eudraguard Biotic), E2 (α-tocopherol and CAP), E3 (α-tocopherol and WPs). Scale bar size 50 μm.

**Figure 4 pharmaceutics-14-01480-f004:**
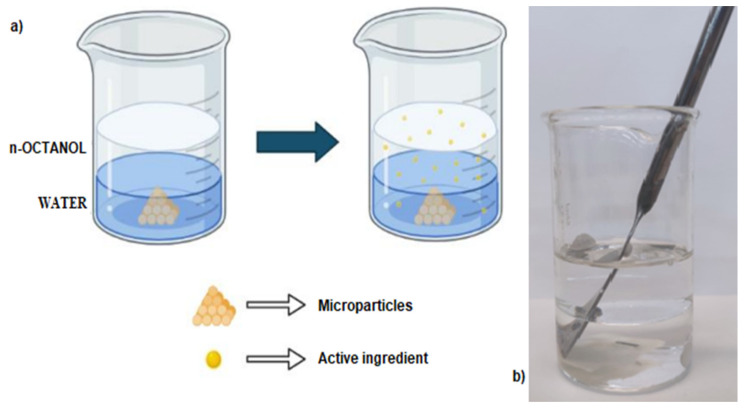
Scheme of the system used for the dissolution/distribution study of the microparticles (created with BioRender.com, accessed on 1 April 2022) (**a**) and photo of the model used experimentally (**b**).

**Figure 5 pharmaceutics-14-01480-f005:**
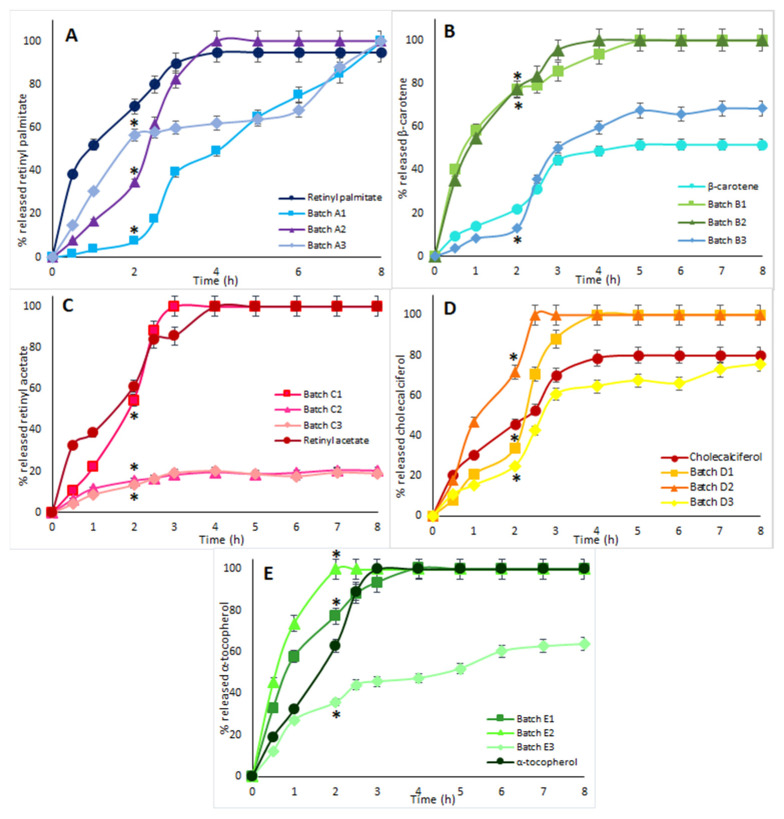
Graphs of the percentage of vitamin released as a function of time: (**A**) retinyl palmitate, batches A1, A2, A3 and pure retinyl palmitate; (**B**) β-carotene, batches B1, B2, B3 and pure β-carotene; (**C**) retinyl acetate, batches C1, C2, C3 and pure retinyl acetate; (**D**) cholecalciferol, batches D1, D2, D3 and pure cholecalciferol; (**E**) α-tocopherol, batches E1, E2, E3 and pure α-tocopherol (mean ± SD, *n* = 3). *p* < 0.05 (*****) for all tested batches versus pure vitamins after 2 h.

**Figure 6 pharmaceutics-14-01480-f006:**
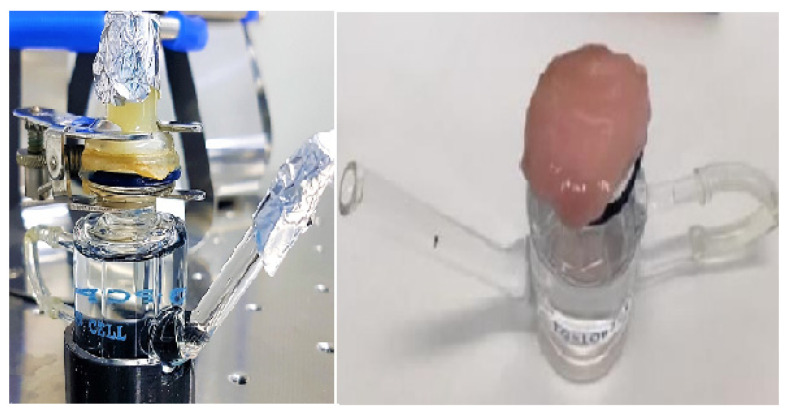
Vertical diffusion of Franz’s cell during the experiment (**left** panel). Positioning of the porcine intestinal mucosa on the acceptor compartment of Franz’s cell (**right** panel).

**Figure 7 pharmaceutics-14-01480-f007:**
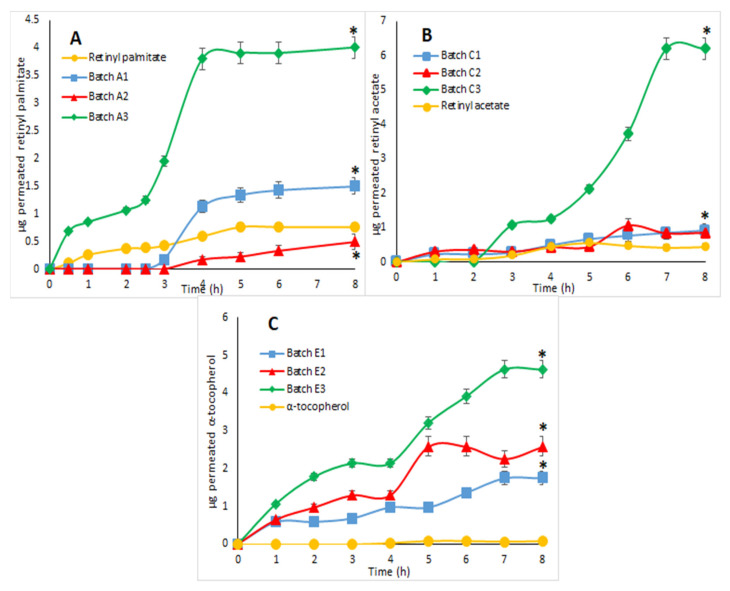
Amount (μg) of permeated vitamin as a function of time: (**A**) retinyl palmitate, batches A1, A2, A3 and pure retinyl palmitate; (**B**) retinyl acetate, batches C1, C2, C3 and pure retinyl acetate; (**C**) α-tocopherol, batches E1, E2, E3 and pure α-tocopherol (mean ± SD, *n* = 3). *p* < 0.05 (*) for all tested batches versus pure vitamins after 8 h.

**Table 1 pharmaceutics-14-01480-t001:** Summary of yield, DL and EE ± SD (mean ± SD, *n* = 3) values for microparticle batches: A1 (retinyl palmitate and Eudraguard Biotic), A2 (retinyl palmitate and CAP), A3 (retinyl palmitate and WPs), B1 (β-carotene and Eudraguard Biotic), B2 (β-carotene and CAP), B3 (β-carotene and WPs), C1 (retinyl acetate and Eudraguard Biotic), C2 (retinyl acetate and CAP), C3 (retinyl acetate and WPs), D1 (cholecalciferol and Eudraguard Biotic), D2 (cholecalciferol and CAP), D3 (cholecalciferol and WPs), E1 (α-tocopherol and Eudraguard Biotic), E2 (α-tocopherol and CAP), E3 (α-tocopherol and WPs). The data are statistically different, with the exception of A1 compared to E2, A2 compared to D2, C3 compared to E3, for the yield; E1 compared to C2, A3 compared to B3, A2 compared to D3 and B2 compared to C3, for the DL; C1 compared to A3, for the EE.

Batches	Yield% ± SD	DL% ± SD	EE% ± SD
A1	52.8 ± 0.2	3.1 ± 0.2	34.2 ± 0.2
B1	63.9 ± 0.3	3.5 ± 0.3	41.3 ± 0.3
C1	43.2 ± 0.1	1.6 ± 0.1	17.1 ± 0.4
D1	56.7 ± 0.1	1.4 ± 0.1	77.7 ± 0.1
E1	47.3 ± 0.4	0.6 ± 0.1	25.9 ± 0.2
A2	45.1 ± 0.2	0.5 ± 0.4	31.8 ± 0.2
B2	66.2 ± 0.1	1.8 ± 0.2	20.2 ± 0.3
C2	46.9 ± 0.5	0.6 ± 0.3	43.3 ± 0.5
D2	45.2 ± 0.3	1.6 ± 0.2	83.6 ± 0.3
E2	52.7 ± 0.2	4.2 ± 0.3	71.9 ± 0.2
A3	57.9 ± 0.2	1.1 ± 0.1	17.1 ± 0.1
B3	65.2 ± 0.1	1.1 ± 0.2	18.1 ± 0.1
C3	44.1 ± 0.1	1.8 ± 0.4	30.2 ± 0.4
D3	48.3 ± 0.3	0.5 ± 0.5	39.8 ± 0.3
E3	44.2 ± 0.4	2.4 ± 0.3	81.7 ± 0.5

**Table 2 pharmaceutics-14-01480-t002:** Average diameter (μm) ± SD (mean ± SD, *n* = 3) for each batch of microparticles (A1, B1, C1, D1, E1, A2, B2, C2, D2, E2, A3, B3, C3, D3, E3).

Batches	Average Diameter (μm) ± SD
A1	10.4 ± 2.3
B1	12.1 ± 3.3
C1	8.8 ± 3.2
D1	13.6 ± 3.7
E1	10.8 ± 4.4
A2	11.5 ± 3.4
B2	8.9 ± 2.3
C2	9.2 ± 4.5
D2	12.2 ± 2.9
E2	10.6 ± 4.1
A3	14.9 ± 4.3
B3	6.7 ± 2.5
C3	11.8 ± 3.9
D3	10.1 ± 2.5
E3	10.7 ± 4.9

**Table 4 pharmaceutics-14-01480-t004:** Mass variations (∆m) as mg and percentage ± SD (mean ± SD, *n* = 3), calculated by thermogravimetric analysis of all microparticle batch samples (A1, A2, A3, B1, B2, B3, C1, C2, C3, D1, D2, D3, E1, E2, E3), carried out 6 months after their production.

Batches	Δm (mg) ± SD	Δm (%) ± SD
A1	0.481 ± 0.153	3.12 ± 0.54
A2	0.453 ± 0.165	3.41 ± 0.42
A3	0.411 ± 0.189	3.22 ± 0.32
B1	0.347 ± 0.254	2.57 ± 0.71
B2	0.283 ± 0.145	2.23 ± 0.69
B3	0.335 ± 0.187	2.42 ± 0.85
C1	0.355 ± 0.211	2.91 ± 0.33
C2	0.468 ± 0.133	3.71 ± 0.25
C3	0.463 ± 0.147	3.89 ± 0.27
D1	0.461 ± 0.188	3.29 ± 0.22
D2	0.511 ± 0.221	3.78 ± 0.31
D3	0.481 ± 0.198	3.45 ± 0.46
E1	0.394 ± 0.155	3.39 ± 0.55
E2	0.503 ± 0.132	3.98 ± 0.41
E3	0.386 ± 0.119	3.78 ± 0.39

## Data Availability

Not applicable.

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
