# Peer review of "Gastro-Resistant Microparticles Produced by Spray-Drying as Controlled Release Systems for Liposoluble Vitamins"

_pharmaceutics, 2022, doi:10.3390/pharmaceutics14071480_

Round 1
Reviewer 1 Report
This paper used spray drying technique to prepare gastro-resistant microparticles for controlled release of liposoluble vitamins. The experiments are rigorously articulated but the writing and grammar needs significant improvement.
Lines 20-25 the authors need to focus more on briefly discussing the results and major findings rather than describing what experiments and characterizations were performed.
Line 49 ‘taken us’ should be ‘taken up’
Lines 55-82 The information about different vitamins need to be simplified. The authors need to focus more on discussing why the Gastro-resistant microparticles is essential for uptake of these nutrients.
Lines 238-241 the authors need to discuss how the yield and drug loading rate reported in this study compare with state-of-the-art processes.
Lines 246-247 A detailed size distribution analysis for each sample is needed.
Lines 307-319 A more representative aqueous phase such as FaSSIF (Fasted State Simulated Intestinal Fluid) is needed to demonstrate the Gastro-resistant property of MPs prepared in this study.
Statistical analysis is necessary for data in Fig 4 and Fig 6.
Author Response
Dear Editor,
the authors thank you for your email regarding the referee’s reports for manuscript ID: pharmaceutics-1767757 entitled " Gastro-resistant microparticles produced by spray-drying as controlled release systems for liposoluble vitamins" and for giving us the possibility of improving the quality of the manuscript.
Please find in the following the detailed responses to the Referees’ comments.
The changes were added in the revised manuscript as red text.
Reviewer 1
Comments and Suggestions for Authors
This paper used spray drying technique to prepare gastro-resistant microparticles for controlled release of liposoluble vitamins. The experiments are rigorously articulated but the writing and grammar needs significant improvement.
The authors thank the reviewer for the overall positive comments. According to the reviewer’s suggestion, the text was carefully re-read and English errors were corrected when found.
- Lines 20-25 the authors need to focus more on briefly discussing the results and major findings rather than describing what experiments and characterizations were performed.
According to the reviewer’s comment, the authors modified the abstract to include more numerical data regarding the outcomes of the experimental studies.
- Line 49 ‘taken us’ should be ‘taken up’
According to the reviewer’s comment, the correction was reported in the article as red text.
- Lines 55-82 The information about different vitamins need to be simplified. The authors need to focus more on discussing why the Gastro-resistant microparticles is essential for the uptake of these nutrients.
Accordingly, the first part of the introduction has been revised and edited (lines 65-83 of the revised version).
- Lines 238-241 the authors need to discuss how the yield and drug loading rate reported in this study compare with state-of-the-art processes.
The authors have reported the right bibliographic references to the current state of the art. In this regard, the following new text and references were added in the revised manuscript, pag.6, lines 253-255:
“These results obtained and reported here are completely in line with the current state of the art of this new technique of microparticle production by spray-drying [21] [23] [24].”
6. Lines 246-247 A detailed size distribution analysis for each sample is needed.
According to the reviewer’s comment, a more detailed size distribution analysis has been added in the table 2 and the following new text at page 7-8 of the revised version:
“In particular, batches 1 (average diameter between 9 and 14 mm) and 2 (average diameter between 9 and 12 mm) of the different active ingredients appear to have a more spherical homogeneous morphology and a smaller diameter average size range, probably due to the presence of different excipients, such as mannitol, gum arabic, eudraguard biotic which improve their surface characteristics. On the other hand, batches 3 (average diameter between 7 and 15 mm) have a more non-homogeneous morphology and have several col-lapsed microparticles, probably due to the lack of excipients capable of guaranteeing these characteristics. In spite of these differences, in any case, the vitamins are encapsulated within the MPs and no correlation has been noted between these surface characteristics and the dissolution and/or permeation profiles of these systems.”
7. Lines 307-319 A more representative aqueous phase such as FaSSIF (Fasted State Simulated Intestinal Fluid) is needed to demonstrate the Gastro-resistant property of MPs prepared in this study.
The authors thank the reviewer for the suggestion, however, the choice of this aqueous phase was not accidental but is supported by studies, cited in the present work, such as:
- Caruana R., et al., “Enhanced anticancer effect of quercetin microparticles formulation obtained by spray drying,” International Journal of Food Science and Technology, 2022, doi: 10.1111/ijfs.15539.
-Shirzad Azarmi; Wilson Roa; Raimar Löbenberg (2007). Current perspectives in dissolution testing of conventional and novel dosage forms. , 328(1), 12–21. doi:10.1016/j.ijpharm.2006.10.001
-Villena, María José Martín; Lara-Villoslada, Ferderico; Martínez, María Adolfina Ruiz; Hernández, María Encarnación Morales (2015). Development of gastro-resistant tablets for the protection and intestinal delivery of Lactobacillus fermentum CECT 5716. International Journal of Pharmaceutics, 487(1-2), 314–319. doi:10.1016/j.ijpharm.2015.03.078.
It is the author’s opinion that the aqueous media used in this study is sufficiently predictive to demonstrate the potential pH-sensitive release profile. It is the pH of the media the most important parameter affecting the gastro-resistant properties, also in accordance with the current Official Pharmacopeia.
- Statistical analysis is necessary for data in Fig 4 and Fig 6.
As reported in the methods section, all results obtained were expressed as mean ± standard deviation (SD). Statistical differences were analyzed by Student's T-Test and P-value <0.05 was defined as the level of statistical significance. For greater accuracy, the authors have also included this information in the captions of figures 5 and 7 of the revised manuscript.
Reviewer 2 Report
Dear Authors,
During reviewing the manuscript I found a few problems.
1. How did you calculate the Yield?
2. Formula for %DL and %EE - "mass of active" - do you mean active substances?
3. "In vitro dissolution studies" - Did you check the solubility in organic phase? (sink condition); "One ml sample of the upper organic phase (n-octanol) 201 was taken at regular time intervals, immediately read with the spectrophotometer UV 202 Jasco V-760 (cuvette 1ml, optical path 10mm) and poured into the beaker." - this phrase should be written at the end of this part.
4. "Ex-vivo permeation studies" - What amount of fluid did you take for analysis? What kind of method did you use for analysis? If spectrophotometry, it could be unsufficient sensitivity.
Author Response
Dear Editor,
the authors thank you for your email regarding the referee’s reports for manuscript ID: pharmaceutics-1767757 entitled " Gastro-resistant microparticles produced by spray-drying as controlled release systems for liposoluble vitamins" and for giving us the possibility of improving the quality of the manuscript.
Please find in the following the detailed responses to the Referees’ comments.
The changes were added in the revised manuscript as red text.
Reviewer 2
Comments and Suggestions for Authors
Dear Authors, During reviewing the manuscript I found a few problems.
- How did you calculate the Yield?
The authors added the formula for calculating the process yield in the methods section, as suggested by the reviewer.
- Formula for %DL and %EE - "mass of active" - do you mean active substances?
Yes, the authors indicate the mass of the active ingredient. In order to avoid misunderstandings, the missing word “ingredient” has been inserted in the two mathematical formulas of DL% and EE%.
- "In vitro dissolution studies" - Did you check the solubility in organic phase? (sink condition); "One ml sample of the upper organic phase (n-octanol) 201 was taken at regular time intervals, immediately read with the spectrophotometer UV 202 Jasco V-760 (cuvette 1ml, optical path 10mm) and poured into the beaker." - this phrase should be written at the end of this part.
The authors checked the solubility of the different active ingredients in n-octanol, resulting the appropriate solvent for these lipophilic vitamins (confirmed by the bibliography and by our experimental tests). According to the reviewer’s comment, the quoted sentence has been moved to the end of the paragraph.
- "Ex-vivo permeation studies" - What amount of fluid did you take for analysis? What kind of method did you use for analysis? If spectrophotometry, it could be unsufficient sensitivity.
According to the reviewer’s suggestion, the authors specified in the Material and Methods section (lines 221-224 of the revised manuscript): “ 200 μL samples, from the acceptor compartment, were picked up every 60 minutes for eight hours, immediately diluted up to 1 ml with n-octanol and read with the spectrophotometer UV Jasco V-760 (cuvette 1ml, optical path 10mm)”. In addition, the authors confirm that the spectroscopic analysis method was found to be sufficiently sensitive for the detection of the active ingredients discussed in this article.
Reviewer 3 Report
I have reviewed through the whole manuscript critically and I found that authors have attempted a good work about using the gastro-resistant microparticles via spray-drying as controlled release systems for liposoluble vitamins. However, a major revision is required due to multiple issues to be addressed. Below please find my comments for your consideration, thanks.
1. Several minor typos:
Line 45, delete “more”;
Line 49, “taken us as” could be “taken as”;
Line 125, “advantages”;
Line 140, “cm-1”;
Line 163, 0.1 g cholecalciferol, “or” 0.25 g α-tocopherol;
Line 285, delete “which”;
Line 315, “in vitro model”.
2. Format optimization and being concise:
Line 48-59, it is better to combine these 3 paragraphs into one whole concise paragraph because all contents are talking about vitamin A;
Line 60-75 and line 76-83, same issues for vitamin D and vitamin E;
Line 120-122, it is better to combine this short paragraph with previous long paragraph in line 108-119;
Line 151, there could be one blank line before “Preparation of the gastro-resistant microparticle formulations”; please format it as same as other sub-titles, e.g. delete blank line 183 for “Drug loading evaluation”.
3. Literature review:
Line 48-83, please add solubility data for all 5 vitamins used in this study at specific pH (e.g. gastric acidity pH 1-2, intestinal region pH 5.0-6.8). This manuscript focused on how to achieve controlled release of fat-soluble vitamins through spray dried MPs, not on the nutrient essentials so that the authors should simplify the nutrient functionality of each vitamin and highlight the needs of enhanced bioavailability of these lipophilic vitamins through advanced MP technology.
Line 155, in regards to high inlet T at 130 °C, in the introduction section please add please add thermal stability data of all Vitamins and 3 carrier excipients (e.g. glass transition point, melting point) used in spray drying process.
4. Line 117-118, please explain whether whey proteins (WPs) are soluble in water over a wide range of pH (e.g. from pH 2 to pH 9)? What is the rational to choose WPs as food-derived gastro-resistant carriers for controlled release?
5. Line 157, please indicate which components were insoluble and which components were soluble in each suspension system; What is the particle size distribution (PSD) of each suspension?
6. Line 160, please describe the functionality of each excipients added, including Arabic gum, mannitol and soy lecithin.
7. Line 165, please indicate the pH of 30% w/v solution of Eudraguard Biotic and final pH of final suspension for spray drying.
8. Line 176, please indicate the extract components in the whole milk used in the study.
9. Line 213, please indicate extract number (mL) for each volume withdrawn for sampling.
10. Line 232-233, please add detailed optimization work in the supplementary files.
11. Line 248, for Figure 2, please highlight the scale bar in color.
12. Line 258-263, please use “the vitamins” to replace “drugs”.
13. Line 287, the excipient Eudraguard Biotic (glass transition point is around 50 °C) should be still in a viscous or rubbery state as Outlet T: 70-73 °C, so that the produced microparticles could be sticky to the inner wall of drying chamber. Please explain how to successfully collect the solid dispersions sample?
14. Major issue in Line 319, for Figure 3 (b) and line 342 Figure 4, the white block in the beaker is the paper sachet? Is it in the octanol phase or water phase in the beaker? From my viewpoint, it was in the organic phase rather than the water phase. Please explain how the Vitamins in paper sachets were finally detected in the organic octanol phase? Were they first supposed to be dissolved in water phase and then transitioned to the organic octanol phase? Or were they directly dissolved in octanol phase? For example, for Figure 4 D, is cholecalciferol soluble in water at pH 1 and pH 6.8? How could it reach almost 100% release after 1 hour if this lipophilic compound is not soluble in water phase? I am afraid the cholecalciferol in paper sachets was directly released into the organic octanol phase as most parts of paper sachet were immersed in organic phase rather than water phase as shown in the Figure 3 (b). The authors should improve the in vitro biphasic dissolution study design. The current dissolution results in Figure 4 indicated that 3 different gastro-resistant MPs only worked for each specific vitamin, e.g. A1 achieved controlled release but B1, C1 and E1 all failed to obtain a controlled release profile. The authors must revise the corresponding abstract content and conclusions considering that gastro-resistant excipient seems to be selective for each lipophilic vitamin, and more in-depth study and discussions (probably explain why A1 worked while others failed based on FTIR and DSC data interpretation) are needed behind this phenomenon if any future consideration, thanks!
Author Response
Dear Editor,
the authors thank you for your email regarding the referee’s reports for manuscript ID: pharmaceutics-1767757 entitled " Gastro-resistant microparticles produced by spray-drying as controlled release systems for liposoluble vitamins" and for giving us the possibility of improving the quality of the manuscript.
Please find in the following the detailed responses to the Referees’ comments.
The changes were added in the revised manuscript as red text.
Comments and Suggestions for Authors
I have reviewed through the whole manuscript critically and I found that authors have attempted a good work about using the gastro-resistant microparticles via spray-drying as controlled release systems for liposoluble vitamins. However, a major revision is required due to multiple issues to be addressed. Below please find my comments for your consideration, thanks.
Authors thank the reviewer for the overall positive comments and for giving us the possibility of improving the quality of the manuscript.
- Several minor typos:
Line 45, delete “more”;
Line 49, “taken us as” could be “taken as”;
Line 125, “advantages”;
Line 140, “cm-1”;
Line 163, 0.1 g cholecalciferol, “or” 0.25 g α-tocopherol;
Line 285, delete “which”;
Line 315, “in vitro model”.
According to the suggestions of the reviewer, all the above changes were made in the revised manuscript and highlighted as red text.
- Format optimization and being concise:
Line 48-59, it is better to combine these 3 paragraphs into one whole concise paragraph because all contents are talking about vitamin A;
Line 60-75 and line 76-83, same issues for vitamin D and vitamin E;
Line 120-122, it is better to combine this short paragraph with previous long paragraph in line 108-119;
Line 151, there could be one blank line before “Preparation of the gastro-resistant microparticle formulations”; please format it as same as other sub-titles, e.g. delete blank line 183 for “Drug loading evaluation”.
The authors thank the reviewer for the suggestions. Accordingly, the introduction section was simplified as a suggestion.
- Literature review:
Line 48-83, please add solubility data for all 5 vitamins used in this study at specific pH (e.g. gastric acidity pH 1-2, intestinal region pH 5.0-6.8). This manuscript focused on how to achieve controlled release of fat-soluble vitamins through spray dried MPs, not on the nutrient essentials so that the authors should simplify the nutrient functionality of each vitamin and highlight the needs of enhanced bioavailability of these lipophilic vitamins through advanced MP technology.
According to the reviewer's comment, the section of the introduction regarding the functionality of the nutrients analyzed in this study has been summarized. The solubility of these active ingredients (regardless of pH) in aqueous solutions is negligible; they are all extremely hydrophobic substances. Precisely for this reason, the purpose of this work is to improve the dissolution and permeation of the active ingredient according to a pH-dependent profile thanks to the three different formulations we have created, as it is well explained in the Introduction section of the revised manuscript:
“Nowadays, it is very common to consume such vitamins as supplements, despite the efficacy of the above lipophilic vitamin being often compromised by their poor water sol-ubility, rapid degradation, and low absorption, resulting in low bioavailability. To over-come these problems, researchers have focused on promising delivery technologies, such as microparticles, which allow to improve the dissolution and permeation of these active ingredients according to a gastro-resistant profile. On the other hand, controlled vitamins release may reduce potential hypervitaminosis syndromes, side effects or frequent ad-ministration [12].”
Line 155, in regards to high inlet T at 130 °C, in the introduction section please add please add thermal stability data of all Vitamins and 3 carrier excipients (e.g. glass transition point, melting point) used in spray drying process.
The authors specify that the supplementary information shows the images of the graphs of the DSC analysis of the MPs, excipients and active ingredients, where the individual melting points have been highlighted.
Line 176, please indicate the extract components in the whole milk used in the study.
Line 248, for Figure 2, please highlight the scale bar in color.
Line 160, please describe the functionality of each excipients added, including Arabic gum, mannitol and soy lecithin.
Line 213, please indicate extract number (mL) for each volume withdrawn for sampling.
Line 258-263, please use “the vitamins” to replace “drugs”.
The authors agreed to all suggestions of the reviewer and the text of the revised manuscript was implemented accordingly.
Line 287, the excipient Eudraguard Biotic (glass transition point is around 50 °C) should be still in a viscous or rubbery state as Outlet T: 70-73 °C, so that the produced microparticles could be sticky to the inner wall of drying chamber. Please explain how to successfully collect the solid dispersions sample?
Actually, the production method of microparticles by the spray-drying technique allows, starting from a solution/suspension, to obtain a dry and stable microparticle powder. All the advantages of this microparticle production technique are described in the articles by Modica de Mohac, L. et al. and Caruana, R. et al., cited in the manuscript. Furthermore, in this study, the final product is free-flowing (not sticky), after spray drying process, thanks to the appropriately chosen co-formulants. In this regard, the photo of some microparticle samples obtained by the spray-drying technique was added in the revised manuscript as Figure 2.
Line 117-118, please explain whether whey proteins (WPs) are soluble in water over a wide range of pH (e.g. from pH 2 to pH 9)? What is the rational to choose WPs as food-derived gastro-resistant carriers for controlled release?
WPs are poorly soluble in an acidic medium. Their solubility increases with pH. This is the reason why they are good candidates as a gastro-resistant excipient (see references 17 and 18 of the revised manuscript and the following as well:
- Wijaya, Wahyu; Harfieyanto, Rachel Catherina; Dewettinck, Koen; Patel, Ashok R.; Van der Meeren, Paul (2019). Whey protein isolate-low methoxyl pectin nanocomplexes improve physicochemical and stability properties of quercetin in a model fat-free beverage. Food & Function, doi: 10.1039.c8Fo02350f.).
Moreover the presence of β-lactoglobulin justify the WPs ability to complex hydrophobic molecules, thus increasing their water solubility and acting as potential colloidal vector, as already reported:
- Patricia Zimet, Yoav D. Livney, Beta-lactoglobulin and its nanocomplexes with pectin as vehicles for ω-3 polyunsaturated fatty acids, Food Hydrocolloids, Volume 23, Issue 4,2009, Pages 1120-1126, https://doi.org/10.1016/j.foodhyd.2008.10.008.
Line 157, please indicate which components were insoluble and which components were soluble in each suspension system; What is the particle size distribution (PSD) of each suspension?
Actually, the principal insoluble components of the spray-drying mixture are the lipophilic vitamins. However, the presence of emulsifying agents added in the formulations before spray-drying (soy lecithin and arabic gum for batches 1 e 2, and WPs for batches 3) improve the dispersibility of the lipophilic component and the stability of the suspension system. Feed dispersion analysis by DLS demonstrated that particle size distribution (PSD) of each suspension is below 1 mm.
Line 165, please indicate the pH of 30% w/v solution of Eudraguard Biotic and final pH of final suspension for spray drying.
The pH of 30% w/v solution of Eudraguard Biotic is around 2.5 and the pH of final suspension for spray drying is around 4.5.
Line 232-233, please add detailed optimization work in the supplementary files.
The optimization work required months of preliminary attempts and drying tests in order to define the optimal relative quantities of every single component of the feed dispersions, as well as the characterization analyses of the produced batches, before proceeding with any changes in the parameters. Therefore, it is the opinion of the authors that adding details of the optimization work is not relevant to readers for better understand the aims of this study.
- Major issue
in Line 319, for Figure 3 (b) and line 342 Figure 4, the white block in the beaker is the paper sachet? Is it in the octanol phase or water phase in the beaker? From my viewpoint, it was in the organic phase rather than the water phase. Please explain how the Vitamins in paper sachets were finally detected in the organic octanol phase? Were they first supposed to be dissolved in water phase and then transitioned to the organic octanol phase? Or were they directly dissolved in octanol phase? For example, for Figure 4 D, is cholecalciferol soluble in water at pH 1 and pH 6.8? How could it reach almost 100% release after 1 hour if this lipophilic compound is not soluble in water phase? I am afraid the cholecalciferol in paper sachets was directly released into the organic octanol phase as most parts of paper sachet were immersed in organic phase rather than water phase as shown in the Figure 3 (b). The authors should improve the in vitro biphasic dissolution study design. The current dissolution results in Figure 4 indicated that 3 different gastro-resistant MPs only worked for each specific vitamin, e.g. A1 achieved controlled release but B1, C1 and E1 all failed to obtain a controlled release profile. The authors must revise the corresponding abstract content and conclusions considering that gastro-resistant excipient seems to be selective for each lipophilic vitamin, and more in-depth study and discussions (probably explain why A1 worked while others failed based on FTIR and DSC data interpretation) are needed behind this phenomenon if any future consideration, thanks!
In order to clarify to the reviewer the misunderstanding raised by the photo in Figure 3 b ( Figure 4 b in the revised manuscript) it is important to underline that the photo of the system used during the experiment in the article (figure 4b) was taken at the end of the experiment when the restraint system that prevented the paper sachet, containing the microparticles or the active ingredient, from passing into the n-octanol phase, was already been removed. In this way, it was possible to allow the paper sachet to remain exclusively in the aqueous phase throughout the experiment. The authors insert a new photo, in the revised manuscript, of the real system used during the experiment.
Actually, the experimental model used during the dissolution study is a biphasic system consisting of a 1:1 ratio aqueous/organic mixture. For the first two hours of the experiment, the aqueous phase had a pH equal to 1 and only after 2 hours, the pH was increased to mimic the enteric passage. The choice of this system is justified by the authors' desire to mimic the aqueous gastric fluid in which the active ingredient is released and an organic interface (n-octanol) where the active ingredients are very soluble, thus guaranteeing sink conditions in the aqueous phase. (Reference article: S. Azarmi, W. Roa, and R. Löbenberg, “Current perspectives in dissolution testing of conventional and novel dosage forms,” International Journal of Pharmaceutics, vol. 328, no. 1 SPEC. ISS. Elsevier, pp. 12–21, Jan. 02, 2007. doi: 10.1016/j.ijpharm.2006.10.001).
As regards the dissolution profile of cholecalciferol, the authors also agree with this anomalous result and thank the reviewer for having noted it. Therefore, the experimental test was repeated with accuracy and the new result revealed the correct release profile of cholecalciferol. Figure 5 D was appropriately corrected.
Finally, the abstract was properly revised as well as also the Discussion section.
Reviewer 4 Report
The authors did not provide a powerful discussion of the results with previous literature since the authors only focus on describing their results. The scientific quality of the manuscript is insufficient for publication in its current form. Specific questions and points requiring attention are itemized below.
1. Reviewer’s comment: The introduction was poorly written. It requires additional information on previous attempts when similar materials were used and what were the results.
2. Reviewer’s comment: It is difficult to understand the novelty of the work, also the authors do not previously introduce us to the most important antecedents.
3. Reviewer´s comment: What is the innovation of this paper? What is new in this work? It is not clear.
4. Review comment: (line 31, page 1). ¿What is the function of the vitamins in the body? Explain more in detail its biological mechanisms.
5. Reviewer comments: The authors must define “microencapsulation” in the introduction section. The authors can cite the next article: 10.1080/00914037.2022.2042289.
6. Reviewer´s comment: (line 90, page 2). The authors must describe the wall materials more commonly used to encapsulate vitamin liposoluble.
7. Reviewer´s comment: In the material section, the authors must give more information about polymers, for example., molecular weight, source, software used, etc etc. Explain more in detail.
8. Reviewer´s comment: In SEM analysis, the authors must give more information of the preparation of the sample, magnification, coating process, etc etc..
9. Reviewer´s comment: DSC in Celsius degree.
10. Reviewer´s comment: (line 144, page 3). “120 μl aluminum crucibles”, hermetic or non-hermetics???
11. Reviewer´s comment: In Thermogravimetric analysis (TGA), the authors must homogenize the units of temperature… Celsius or kelvin???
12. Reviewer´s comment: In the section preparation of microparticles, the authors can do a table with each formulation. Since the procedure is not very clear.
13. Reviewer´s comment: The authors can add images of the powders obtained.
14. Reviewer´s comment: Table 1 and Table 2 must show if exist significant differences. The authors can use letters.
15. Reviewer’s comment: The results and discussion sections are poor. More comparison with previous literature should be discussed.
16. Reviewer’s comment: The authors must explain the purpose of each analysis used in the discussion section.
17. Reviewer’s comment: The image quality of Figure 2 is poor. The auhors must show a higher magnification to observe the microstructure of the particles.
18. Reviewer’s comment: “SEM images (Figure 2) showed a quite homogeneous spherical shape of all microparticles batches and an average size ranging between 6 and 20 μm”: How did you measure the average size of the particles? How did you confirm your spherical shape?. This conclusions are not proven.
19. Reviewer’s comment: “DSC analysis was used to evaluate if the production process of microparticles allows the creation of chemical-physical interactions between the excipients and the active ingredient used in each batch produced, upon the spray-drying technique”….. What thermal transitions allow you to confirm those assertions?
20. Reviewer’s comment: In all the comparisons of DSC graphs, it was noticed that the typical melting/degradation peaks of each pure vitamin”. What temperature?? Why do the authors describe it as degradation? Can the authors demonstrate this asseveration?
21. Reviewer’s comment: The authors report an excessive number of graphics, which are not adequately described and discussed. I suggest that the authors show one and two DSC graphics on the manuscript, with the results more critical than responses to the research question.
22. Reviewer’s comment: What is the reason for the different values of fusion enthalpies of the exothermic peaks?.
23. Reviewer’s comment: The authors must add the TGA thermogram and explain each mass loss, not just water loss between 25 and 550 °C.
24. Reviewer’s comment: The authors must evaluate the hygroscopicity and water activity of the microparticles.
25. Reviewer’s comment: In the release profile, the authors must fit mathematical models that explain the results.
Author Response
Dear Editor,
the authors thank you for your email regarding the referee’s reports for manuscript ID: pharmaceutics-1767757 entitled " Gastro-resistant microparticles produced by spray-drying as controlled release systems for liposoluble vitamins" and for giving us the possibility of improving the quality of the manuscript.
Please find in the following the detailed responses to the Referees’ comments.
The changes were added in the revised manuscript as red text.
Comments and Suggestions for Authors
The authors did not provide a powerful discussion of the results with previous literature since the authors only focus on describing their results. The scientific quality of the manuscript is insufficient for publication in its current form. Specific questions and points requiring attention are itemized below.
- Reviewer’s comment: The introduction was poorly written. It requires additional information on previous attempts when similar materials were used and what were the results.
The Introduction was properly edited and it is the authors’ opinion that references regarding previous attempts with similar materials were cited, for example, 12-16-17-18-19-20-34-35 references.
- Reviewer’s comment: It is difficult to understand the novelty of the work, also the authors do not previously introduce us to the most important antecedents.
- Reviewer´s comment:What is the innovation of this paper? What is new in this work? It is not clear.
The novelty of the work is properly discussed in the Introduction and in the Conclusions sections.
- Review comment: (line 31, page 1). ¿What is the function of the vitamins in the body? Explain more in detail its biological mechanisms.
The function of the vitamins in the body was sufficiently described in the Introduction section (lines 38-83). Explaining the biological mechanism of the vitamins is not the aim of this study, which instead aims to explore new strategies to improve the dissolution and permeation of these active ingredients according to a gastro-resistant profile.
- Reviewer comments: The authors must define “microencapsulation” in the introduction section. The authors can cite the next article: 10.1080/00914037.2022.2042289.
Accordingly, the above article was cited in the revised manuscript.
- Reviewer´s comment: (line 90, page 2). The authors must describe the wall materials more commonly used to encapsulate vitamin liposoluble.
The authors explored the literature in order to find information about the wall materials more commonly used to encapsulate liposoluble vitamins. The number of references is quite negligible, in particular regarding materials with gastro-resistant properties.
- Reviewer´s comment: In the material section, the authors must give more information about polymers, for example., molecular weight, source, software used, etc etc. Explain more in detail.
The polymer used in this work, in particular, CAP and Eudraguard are commercially available materials and thus the above information are patented. The main characteristics can be taken from the technical data sheets.
- Reviewer´s comment:In SEM analysis, the authors must give more information of the preparation of the sample, magnification, coating process, etc etc..
These information are exhaustively reported in the Materials and Method section; SEM samples not required a coating process.
- Reviewer´s comment:DSC in Celsius degree.
DSC graphs are already reported in Celsius degree.
- Reviewer´s comment:(line 144, page 3). “120 μl aluminum crucibles”, hermetic or non-hermetics???
The used aluminium crucibles were non-hermetics, this was added in the Materials and Method section.
- Reviewer´s comment:In Thermogravimetric analysis (TGA), the authors must homogenize the units of temperature… Celsius or kelvin???
The unit of temperature was homogenized in Celsius.
- Reviewer´s comment:In the section preparation of microparticles, the authors can do a table with each formulation. Since the procedure is not very clear.
The table reporting the precise composition of each prepared formulation (Table 1S) was added in the Supplementary Information section.
- Reviewer´s comment: The authors can add images of the powders obtained.
The requested image has been added in the Results and Discussion section as Figure 2.
- Reviewer´s comment: Table 1 and Table 2 must show if exist significant differences. The authors can use letters.
Table 1 summarizes specific characteristics of each prepared microparticles batches such as yield, DL and EE values, while table 2 (table 3 of the revised manuscript) reports the FTIR peak assignment for each vitamin and MPs batches. However, it is author’s opinion that readers can easily identify the differences between the various batches.
- Reviewer’s comment: The results and discussion sections are poor. More comparison with previous literature should be discussed.
- Reviewer’s comment: The authors must explain the purpose of each analysis used in the discussion section.
The authors decided to include most of the bibliographic references in the introduction section of the revised manuscript. However, the results and discussion section has been revised as well.
- Reviewer’s comment: The image quality of Figure 2 is poor. The auhors must show a higher magnification to observe the microstructure of the particles.
The increase in magnification and power of the electron beam causes a superficial alteration of the organic material; as a consequence an alteration of the morphology of the microparticles was observed upon excessive magnification. Therefore, SEM images were recorded using a suitable magnification for this type of microparticles and sufficient to visualize the morphology and average size of the microparticles. Unfortunately, given the large number of the observed samples is not practical to insert more than one image per sample in the manuscript.
- Reviewer’s comment: “SEM images (Figure 2) showed a quite homogeneous spherical shape of all microparticles batches and an average size ranging between 6 and 20 μm”: How did you measure the average size of the particles? How did you confirm your spherical shape?. This conclusions are not proven.
The average diameter (d)±standard deviation (SD) of the microparticles was determined from the mean value of 100 measurements using ImageJ (USA, version 1.46 v). According to the reviewer’s comment, a more detailed size distribution analysis has been added in the table 2 and the following new text at page 7-8 of the revised version:
“In particular, batches 1 (average diameter between 9 and 14 mm) and 2 (average diameter between 9 and 12 mm) of the different active ingredients appear to have a more spherical homogeneous morphology and a smaller diameter average size range, probably due to the presence of different excipients, such as mannitol, gum arabic, eudraguard biotic which improve their surface characteristics. On the other hand, batches 3 (average diameter between 7 and 15 mm) have a more non-homogeneous morphology and have several col-lapsed microparticles, probably due to the lack of excipients capable of guaranteeing these characteristics. In spite of these differences, in any case, the vitamins are encapsulated within the MPs and no correlation has been noted between these surface characteristics and the dissolution and/or permeation profiles of these systems.”
- Reviewer’s comment: “DSC analysis was used to evaluate if the production process of microparticles allows the creation of chemical-physical interactions between the excipients and the active ingredient used in each batch produced, upon the spray-drying technique”….. What thermal transitions allow you to confirm those assertions?
This point is already discussed in the text of the revised manuscript: “In the DSC graphs of figures 7S-21S, the difference in the thermal behavior of each micro-particles batch was compared to the thermal behavior of the simple physical dry mixture of all the components used in the respective formulation, and with the pure component as well. In all the comparison of DSC graphs, it was noticed that the typical melting peaks of each pure vitamin, still visible in the physical mixture, disappeared in the DSC graph of the batch of microparticles produced by spray drying, with the appearance of a new peak at a temperature different from that of the individual components and characteristic only of the microparticles. This suggests that the drying process with the spray dryer generates a solid in which the physical interactions between the excipients and the incorporated vitamin are very strong.”
- Reviewer’s comment: In all the comparisons of DSC graphs, it was noticed that the typical melting/degradation peaks of each pure vitamin”. What temperature?? Why do the authors describe it as degradation? Can the authors demonstrate this asseveration?
In the Results and Discussion section of the revised manuscript, the authors refer to degradation phenomena always in agreement with the scientific literature and proper experience (see for example references number: 25-32-31-33-34-35-17).
- Reviewer’s comment: The authors report an excessive number of graphics, which are not adequately described and discussed. I suggest that the authors show one and two DSC graphics on the manuscript, with the results more critical than responses to the research question.
The authors agree that the number of DSC plots is high, but this is due to the fact that the batches produced and analysed are numerous; and this is why they have been placed in the supplementary information and not in the main text.
- Reviewer’s comment: What is the reason for the different values of fusion enthalpies of the exothermic peaks?.
See the reply of the point 20.
- Reviewer’s comment: The authors must add the TGA thermogram and explain each mass loss, not just water loss between 25 and 550 °C.
All TGA analyses were recorded in the range 25-550 °C. To be exact, the determination of the water content was calculated by the instrument software in the range 20-120 °C, as in usual. For clarity, the following sentence has been added in the revised version of the manuscript (lines 153-154): “The water content was calculated by the instrument software in the range 20-120 °C.”
- Reviewer’s comment: The authors must evaluate the hygroscopicity and water activity of the microparticles.
The evaluation of the hygroscopicity and water activity of the microparticles is not relevant data for this study. However, the authors took care to study the stability of the produced microparticles by re-determining the water content after 6 months. Although this is not a true hygroscopicity test, it still gives valuable information on the ability of materials used to adsorb moisture.
- Reviewer’s comment: In the release profile, the authors must fit mathematical models that explain the results.
It is the authors’ opinion that the release kinetics and mathematical models are useful to study how a specific property (molecular weight, viscosity, erosion, diffusion of drug through the material) of a material, for example, a polymer, may influence the release. In our case, the use of mathematical models is not applicable, since the various batches contain several different functional excipients and have different compositions. Therefore a mathematical model, even if it exists, would not have allowed the comparison of the various formulations with each other. See the reference below:
-Ghosal, Kajal & Chandra, Aniruddha & Rajabalaya, Rajan & Chakraborty, S & Nanda, A. (2012). Mathematical modeling of drug release profiles for modified hydrophobic HPMC based gels. Die Pharmazie. 67. 147-55. 10.1691/ph.2012.1014.
Reviewer 5 Report
Dear authors,
After the review report, I have several comments: you should include numerical data in the abstract; how was realized figure 3, for example - you should mention how was made this type of figures - copyright conditions if it is necessary?!; you should check the entire document - ex vivo, in vitro, in vivo should be Italic; you should clearly define the controls in section 2 and Results section; the authors should include comments in discussions and/or introduction about future valorization of the research. New comments about the role of vitamins in neurodegenerative pathologies as a new pathway to improve brain activity; also, wich is the role of microbiota?, you should realize a correlation between microbiota bioactivity and bioavailability of this functional compounds.
Best regards!
Author Response
Dear Editor,
the authors thank you for your email regarding the referee’s reports for manuscript ID: pharmaceutics-1767757 entitled " Gastro-resistant microparticles produced by spray-drying as controlled release systems for liposoluble vitamins" and for giving us the possibility of improving the quality of the manuscript.
Please find in the following the detailed responses to the Referees’ comments.
The changes were added in the revised manuscript as red text.
Comments and Suggestions for Authors
After the review report, I have several comments:
- You should include numerical data in the abstract;
Accordingly, the authors modified the abstract to include more numerical data regarding the outcomes of the experimental studies.
- the authors should include comments in discussions and/or introduction about future valorization of the research.
According to the reviewer’s suggestion, the authors added the comment below reported, about future valorization of this research, in the final part of the Results and Discussion section (lines 430-435): “ These results encourage further exploration of the potential of WPs as absorption promoters for molecules of pharmaceutical interest, also through in vivo bioavailability studies. In fact, this advantageous feature has already been recently demonstrated with active molecules of natural origin [17]. Furthermore, it will be interesting to understand if specific components of WPs, such as β-lactoglobulins, are directly responsible for these beneficial vehicle properties [37].”
- New comments about the role of vitamins in neurodegenerative pathologies as a new pathway to improve brain activity; also, wich is the role of microbiota?, you should realize a correlation between microbiota bioactivity and bioavailability of this functional compounds.
The authors thank the reviewer for the interesting suggestions. New text (lines 55-62 of revised manuscript) and related references (n. 3-4) have been added, to describe the relationships between vitamins and neurodegenerative diseases and between vitamins and gut microbiota.
- how was realized figure 3, for example - you should mention how was made this type of figures - copyright conditions if it is necessary?!;
The authors created the mentioned figure (figure 4 in the revised manuscript) using a special program. For this reason, the copywriting quote has been added to the caption.
- you should check the entire document - ex vivo, in vitro, in vivo should be Italic; you should clearly define the controls in section 2 and Results section;
The authors have checked the words in the entire document and have corrected in Italic as suggested.
Reviewer 6 Report
In this study, three different gastro-resistant materials were used to prepare vitamins-loaded microspheres by spray drying, and the physicochemical properties and release characteristics of the microspheres were characterized. The microspheres could improve the oral bioavailability of vitamins with pH-dependent solubility, site-selective release, drying characteristics and higher stability. In particular, the microspheres made from natural formula such of WPs and milk could obviously improve the permeability of vitamins. This work is of great practical significance in the field of pharmaceutics, and it is recommended to accept it after minor revision.
1、In line 46, Please give the full name of the abbreviations of GI tract.
2、In the introduction part, a lot of space was given to the introduction of vitamins, but as an article related to the preparation of microspheres, it is suggested that the introduction of vitamins should be simplified as much as possible.
3. Please add a note to the meaning represented of A1 ~ B1 in Table 1, Table 2.
4、According to TEM results, it is found that the appearance of each group of microspheres is somewhat different, such as some clumps and some scattered. Please give the reasons and point out its influence on the subsequent experimental results.
Author Response
Dear Editor,
the authors thank you for your email regarding the referee’s reports for manuscript ID: pharmaceutics-1767757 entitled " Gastro-resistant microparticles produced by spray-drying as controlled release systems for liposoluble vitamins" and for giving us the possibility of improving the quality of the manuscript.
Please find in the following the detailed responses to the Referees’ comments.
The changes were added in the revised manuscript as red text.
In this study, three different gastro-resistant materials were used to prepare vitamins-loaded microspheres by spray drying, and the physicochemical properties and release characteristics of the microspheres were characterized. The microspheres could improve the oral bioavailability of vitamins with pH-dependent solubility, site-selective release, drying characteristics and higher stability. In particular, the microspheres made from natural formula such of WPs and milk could obviously improve the permeability of vitamins. This work is of great practical significance in the field of pharmaceutics, and it is recommended to accept it after minor revision.
The authors thank the reviewer for the overall positive comments and for giving us the possibility of improving the quality of the manuscript.
1、In line 46, Please give the full name of the abbreviations of GI tract.
According to the reviewer’s comment, the correction was reported in the revised manuscript.
2、In the introduction part, a lot of space was given to the introduction of vitamins, but as an article related to the preparation of microspheres, it is suggested that the introduction of vitamins should be simplified as much as possible.
The first part of the introduction has been revised and edited in accordance with the reviewer's comment.
- Please add a note to the meaning represented of A1 ~ B1 in Table 1, Table 2.
According to the reviewer’s suggestion, the authors have added the meaning of the abbreviations of the several batches in the caption of tables 1 and 3 of the revised manuscript.
4、According to TEM results, it is found that the appearance of each group of microspheres is somewhat different, such as some clumps and some scattered. Please give the reasons and point out its influence on the subsequent experimental results.
The authors imagine that the reviewer probably meant SEM, instead of TEM. Anyway, according to the reviewer's comment, the authors have expanded the section of the Results and Discussion regarding SEM analysis, in order to better clarify any correlations with subsequent results.
Round 2
Reviewer 1 Report
Authors still did not add any statistical analysis to Fig 4 and 6. Authors need to mark the statistical analysis results in the figures.
Author Response
Accordingly, the authors added the statistical analysis information in the Figures 5 and 7 of the revised manuscript.
Reviewer 3 Report
The authors have addressed my previous comments. Below please find some new comments for a consideration.
1. Line 18, The latter “was”.
2. Line 189, please delete “.”
3. Format optimization and being concise:
Line 63-71, it is better to combine these 2 paragraphs into one whole concise paragraph in regards to vitamin A;
Line 72-79, it is better to combine these 2 paragraphs into one whole concise paragraph in regards to vitamin D.
4. Line 268-270: please correct 10,4 as “10.4” and do correction for all numbers in the table 2; In addition, please use same decimal for all numbers, like 10.4 ± 2.3.
Author Response
According to the suggestions of the reviewer, all the changes were made in the second revision of the manuscript, and the new text highlighted in red.
Reviewer 4 Report
The scientific quality of the manuscript is insufficient for publication in its current form. Specific questions and points requiring attention are itemized below. In this sense, the authors still do not adequately answer some important questions.
Reviewer´s comment: The abstract has >200 words. The authors should review the “Author Guidelines of the Pharmaceutics journal properly”.
Reviewer´s comment: Again, it is difficult to understand the novelty of the work. The authors do not previously introduce us to the most important antecedents and describe the novelty with respect to what has already been published previously. describe the similar articles and discuss these differences.... or is your article the only one of its kind??????? Please, explain clearly this novelty...
Reviewer´s comment: (line 253, page 6). These results obtained and reported here are completely in line with the current state of the art of this new technique of microparticle production by spray-drying”. Please, describe one and two references. References [21] [23], or [24].
Reviewer´s comment: In Table 1 and Table 4, the authors must add standard deviations and add if exist significant differences in the yield, DL and EE values of microparticles batches results. The authors can add letters o numbers.
Reviewer´s comment: Again… The results and discussion sections are poor. More comparisons with previous literature should be described and discussed.
Reviewer´s comment: (line 315, page 9). This suggests that the drying process with the spray dryer generates solid in which the physical interactions between the excipients and the incorporated vitamin are very strong”. The authors must add references that support it.
Reviewer´s comment: The authors describe that “DSC analysis was used to evaluate if the production process of microparticles allows the creation of chemical-physical interactions between the excipients and the active ingredient used in each batch produced”. In this sense, nowhere in the article is this relationship described since the authors only describe and explain the fundamental thermal transitions.
Reviewer´s comment: (line 324, page 10). “The Eudraguard Biotic and CAP thermograms showed an endothermic peak, respectively, at around 50 °C and 170 °C attributable to the glass transition of the amorphous polymer”. Endothermic and glass transitions are thermal transitions very different. How are they related? Explain more in detail.
Reviewer´s comment: (line 330, page 10). “No unusual peaks of vitamin degradation were observed in the microparticles”. What temperature and melting enthalpy values??
Reviewer´s comment: (line 334, page 10). “All microparticle samples analyzed immediately after the production process showed an average residual percentage of water equal to about 2.5% by weight, as expected”. The values obtained are good, or bad????. What values are suitable???? The authors must describe and discuss the importance of these values on powders.
Author Response
Reviewer 4
- Reviewer´s comment: The abstract has >200 words. The authors should review the “Author Guidelines of the Pharmaceutics journal properly”.
Accordingly, the authors revised the abstract and it now consists of 200 words.
- Reviewer´s comment: Again, it is difficult to understand the novelty of the work. The authors do not previously introduce us to the most important antecedents and describe the novelty with respect to what has already been published previously. describe the similar articles and discuss these differences.... or is your article the only one of its kind??????? Please, explain clearly this novelty...
The authors do not presume to consider this work the only one of its kind. It is certain that this study can be considered innovative in demonstrating that microparticle formulations based on all-natural and/or food components, such as milk proteins and milk itself, with ever better technological properties can be produced using the spray drying technique if compared with synthetic excipients such as CAP and anionic methacrylic copolymer. This innovation opens the way to a new potential market in which even the most common active ingredients could be formulated in this way. However, authors still confirm that many of the important antecedents regarding similar studies were cited in the manuscript in many references, including references 12, 16, 17, 18, 19, 20, 34, 35.
- Reviewer´s comment: (line 253, page 6). These results obtained and reported here are completely in line with the current state of the art of this new technique of microparticle production by spray-drying”. Please, describe one and two references. References [21] [23], or [24].
The authors have better specified the yield values of the process present in the articles cited in that paragraph and modified the comment cited above, in such a way as to better understand the sentence: " in line with the current state of the art”.
- Reviewer´s comment: In Table 1 and Table 4, the authors must add standard deviations and add if exist significant differences in the yield, DL and EE values of microparticles batches results. The authors can add letters o numbers.
The authors added in the caption of tables 1 and 4 standard deviations for all the data. It is not clear what the reviewer means by “The authors can add letters o numbers”. Anyway, the yield, DL and EE data reported in table 1 are only intended to demonstrate that the batches prepared have characteristics in line with the scientific literature. It is not important, as well as difficult, to underline any significant differences of the three calculated values, if we also consider that the systems have different compositions. Regarding the data reported in the table 4, there are no statistically significant differences, in fact in the text on page 10, lines 326-328, we write: “All microparticle samples analyzed immediately after the production process showed an average residual percentage of water equal to about 2.5% by weight, as expected.”.
- Reviewer´s comment: Again… The results and discussion sections are poor. More comparisons with previous literature should be described and discussed.
Accordingly, the authors have already provided, in the revised version, to insert sufficient references regarding the comparisons with previous literature, see for example:
lines 224-226 “Spray-drying is an advanced micronization process improving the supplements formulations with higher drug dissolution, absorption, and therapeutic efficacy when compared to traditional dosage forms [13][15][21].”;
lines 233-234 “Several batches of each micro delivery system were produced to optimize the spray drying process, yields and DL [21] [22].”;
lines 242-246 “ The process’ yield ranged from 43 to 66% and the DLs of the active ingredients ranged between 0.5% and 3.5% as reassumed in table 1. These results agree with the literature regarding the use of this microparticle production technique and it is believed that optimal process yield values are between 30-70%, when using the mini spray dryer Buchi B-290 [21] [23] [24].”;
- Reviewer´s comment: (line 315, page 9). This suggests that the drying process with the spray dryer generates solid in which the physical interactions between the excipients and the incorporated vitamin are very strong”. The authors must add references that support it.
Accordingly, a new reference describing the importance of DSC analyses evaluating interactions and incompatibilities of excipients with API was reported in line 299 as number 25.
- Reviewer´s comment: The authors describe that “DSC analysis was used to evaluate if the production process of microparticles allows the creation of chemical-physical interactions between the excipients and the active ingredient used in each batch produced”. In this sense, nowhere in the article is this relationship described since the authors only describe and explain the fundamental thermal transitions.
This point is already described in the manuscript from lines 299 to 309.
- Reviewer´s comment: (line 324, page 10). “The Eudraguard Biotic and CAP thermograms showed an endothermic peak, respectively, at around 50 °C and 170 °C attributable to the glass transition of the amorphous polymer”. Endothermic and glass transitions are thermal transitions very different. How are they related? Explain more in detail.
The characteristics thermal peaks of the functional excipients used in our study were described only in order to highlight the differences between the microparticles produced with the spray dryer and the pure excipients or their physical mixture with vitamins. It is certainly not the aim of this study to analyze the thermal profile of eudraguard and CAP, because this has already been done (read at this regard references: 35 and 36).
- Reviewer´s comment: (line 330, page 10). “No unusual peaks of vitamin degradation were observed in the microparticles”. What temperature and melting enthalpy values??
Since we have not observed any anomalous peaks, there are no temperature and enthalpy values to be highlighted.
- Reviewer´s comment: (line 334, page 10). “All microparticle samples analyzed immediately after the production process showed an average residual percentage of water equal to about 2.5% by weight, as expected”. The values obtained are good, or bad????. What values are suitable???? The authors must describe and discuss the importance of these values on powders.
The value of about 2,5 % by weight of average residual percentage of water is typical of materials dried with Mini Spray-Dryer Buchi B-290. Anyway, other studies reported similar values, such as Modica de Mohac et al. . The values obtained are good and “expected”, as specified in the manuscript.
Reviewer 5 Report
No other comments.
Author Response
The authors thank the reviewer for considering the manuscript satisfactory to accepted for publication.
Round 3
Reviewer 1 Report
Addressed my concerns
Author Response
Authors thank the reviewer for considering now the manuscript ready to be accepted for publication.
Reviewer 4 Report
The article can be accepted...
Table 1 and Table 4 must displayed if the significances differences between the means
Author Response
According with reviewer's suggestion the analysis of statistical differences was made for values in table 1 and 4. Therefore, the legend of table 1 was implemented as follow: "The data are statistically different, with the exception of A1 compared to E2, A2 compared to D2, C3 compared to E3, for the yield; E1 compared to C2, A3 compared to B3, A2 compared to D3 and B2 compared to C3, for the DL; C1 compared to A3, for the EE." ;
The following new text was added at page 10, lines 335-337 of revised manuscript: "Only batches with b-carotene (B1, B2, B3) have statistically different lower values compared with each of the other vitamins.".